# Distinctive Clinical Features of Portal Hypertension in Children with Portal Vein Thrombosis Following Liver Transplantation

**DOI:** 10.3390/biomedicines13092061

**Published:** 2025-08-24

**Authors:** Naire Sansotta, Angelo Di Giorgio, Mara Marcella Colusso, Marco Salvi, Paolo Marra, Domenico Pinelli, Alessandra Carobbio, Lorenzo D’Antiga

**Affiliations:** 1Paediatric Hepatology Gastroenterology and Transplantation, Hospital Papa Giovanni XXIII, Piazza Oms 1, 24127 Bergamo, Italy; angelo.digiorgio@uniud.it (A.D.G.); ldantiga@asst-pg23.it (L.D.); 2Paediatric Surgery Unit, Hospital Papa Giovanni XXIII, 24127 Bergamo, Italy; mcolusso@asst-pg23.it; 3Department of Pediatrics, University of Milano-Bicocca, 20126 Milan, Italy; m.salvi39@campus.unimib.it; 4Department of Radiology, Hospital Papa Giovanni XXIII, 24127 Bergamo, Italy; pmarra@asst-pg23.it; 5Department of Medicine and Surgery, University of Milano-Bicocca, 20126 Milan, Italy; 6Department of Transplant Surgery, Hospital Papa Giovanni XXIII, 24127 Bergamo, Italy; dpinelli@asst-pg23.it; 7Fondazione per la Ricerca Ospedale di Bergamo ETS, 24127 Bergamo, Italy; alessandra.carobbio@unimore.it; 8Dipartimento di Scienze Mediche e Chirurgiche, Materno-Infantili e dell’Adulto, Università di Modena-Reggio Emilia, 41121 Modena, Italy

**Keywords:** children, portal vein thrombosis, liver transplantation

## Abstract

**Background**: Portal vein thrombosis (PVT) occurs in nearly 8% of pediatric liver transplants (LT), leading to portal hypertension (PH). This study aims to describe the clinical features and management of PVT in children post-transplant (PVTt) compared to those with PVT in native livers (PVTn). **Methods**: All children diagnosed with PVTt between January 2002 and October 2021 were included. The control group comprised pediatric patients with PVTn diagnosed and managed at our center during the same period. **Results**: PVTt was diagnosed in 37 out of 610 children (6%), while 36 children with PVTn were included as controls. At 5-year follow-up, medium-to-large esophageal varices (grade II–III) developed in 15/37 (38%) PVTt patients compared to 23/36 (64%) PVTn patients (*p* = 0.002). Among 11 patients who bled, upper gastrointestinal bleeding occurred in 2/7 (29%) with PVTt, versus 4/4 (100%) PVTn patients (*p* = 0.06). Mean spleen length was 9.3 cm in PVTt versus 7.4 cm in PVTn (*p* = 0.039). Mean platelet count was 76 × 10^3^/L in PVTt versus 93 × 10^3^/L in PVTn (*p* = 0.16). **Conclusions**: Despite more severe PH and marked hypersplenism, children with PVTt have a reduced risk of developing esophageal varices, but an increased risk of bleeding from the lower gastrointestinal tract. This suggests the need for a different surveillance strategy in this patient group. Individualized care is key, mainly in PVTt, where hypersplenism does not correlate with risk of bleeding from esophageal varices.

## 1. Introduction

In the native liver, non-cirrhotic portal vein thrombosis (PVTn) refers to obstruction and cavernomatous transformation of the portal vein following acute thrombosis, typically occurring in critically ill newborns [1]. PVTn has a multifactorial etiology, including both prothrombotic conditions and local vascular factors [2]. Among these, umbilical vein catheterization during the neonatal period is considered the major risk factor. Nonetheless, up to 50% of pediatric cases remain idiopathic [3].

In pediatric liver transplant (LT) recipients, portal vein thrombosis (PVTt) is one of the most frequent vascular complications [4,5,6,7,8,9]. It occurs in up to 12% of children receiving LT, whether with full-size or reduced-size grafts [10], and is especially common in patients with biliary atresia or portal vein hypoplasia. Contributing factors may include small portal vein diameter, surgical technique, vessel misalignment, excessive vessel length, hypercoagulable states, intraoperative thrombus formation, and failed recanalization [11]. When PVTt occurs early (within three weeks post-LT), it may lead to graft failure, significant morbidity, graft loss, and mortality [8]. Late-onset PVTt, by contrast, develops in a functioning graft and often follows a more subtle, subclinical course [5]. In these cases, PVTt is typically well tolerated and characterized by cavernous transformation of the portal vein and the development of hepatopetal collaterals [12].

Portal hypertension (PH) developing after PVT—whether in native or transplanted livers—is a form of non-cirrhotic, presinusoidal, prehepatic PH [13]. Hemodynamic features include reduced blood flow through an abnormal venous network, increased arterial inflow, recurrent thromboses, and blood flow redistribution. In cavernous transformation, multiple dilated, tortuous venous collaterals form in the periportal and subhepatic regions in an attempt to bypass the obstructed extrahepatic portal vein [13,14]. However, this compensatory mechanism is often inadequate to decompress the portal system, leading to the formation of portosystemic collaterals. In this context, the risk of gastrointestinal bleeding is significantly higher and typically occurs earlier than in other forms of portal hypertension [13].

The most common clinical signs of PH include splenomegaly—often measured in standard deviations (SD) from the age-related mean [15]—hypersplenism, esophageal varices, and gastrointestinal bleeding. Up to 80% of children with PVTn experience at least one episode of upper gastrointestinal bleeding, which is associated with considerable morbidity [16,17].

While some risk factors for developing PH after LT (especially secondary to PVTt) are known, there is no validated predictive model that reliably identifies pediatric LT patients who will develop clinically significant PH. Some authors have proposed non-invasive methods for identifying children at risk of esophageal varices, such as the platelet count/spleen length z-score ratio, platelet count alone, and the aspartate aminotransferase-to-platelet ratio index (APRI) [18,19]. These indices have shown utility in cirrhotic PH but lack validation in pediatric PVT populations.

Although several case series have been published on the management of esophagogastric varices in children with PVT, therapeutic approaches vary widely between centers [20,21]. Endoscopic treatments, including sclerotherapy and endoscopic variceal ligation, are effective in controlling varices. A standardized approach has been proposed by the Baveno Pediatric Consensus, which includes pharmacologic, endoscopic, and surgical options such as mesoportal bypass [22,23,24]. For children who are not candidates for mesoportal bypass or who fail endoscopic control, radiologic interventions such as percutaneous angioplasty and/or transjugular intrahepatic portosystemic shunt (TIPS) [25], as well as surgical shunting, may be considered [17,26,27,28]. However, a standardized strategy for PVT occurring post-transplant has not yet been established. Given the non-negligible incidence of PVTt, characterizing its clinical features is important. This study aims to evaluate the clinical presentation of PH in children diagnosed with PVTt at a large tertiary center and to determine whether management strategies used for PVTn can also be applied to the post-transplant population.

## 2. Materials and Methods

This was a retrospective study conducted at a single tertiary care center, including consecutive pediatric (<18 years old) patients with portal vein thrombosis (PVT), either in native livers (PVTn) or in transplanted grafts (PVTt). We retrospectively reviewed clinical data, endoscopic reports, and radiological imaging for all children diagnosed with PVT between January 2002 and October 2021. Inclusion criteria were an imaging diagnosis of PVT and clinical signs of portal hypertension, namely hypersplenism, defined as a platelet count below 150,000/mm^3^. Spleen size was measured via ultrasound and expressed in standard deviations (SD) from the age-related normal mean. The exclusion criterion was an imaging diagnosis of acute or subacute PVT.

Follow-up data included blood tests, primary liver disease, comorbidities, presence of varices, gastrointestinal bleeding episodes, endoscopic treatment, radiologic procedures, and surgical interventions. We also reviewed surgical techniques used during liver transplantation, particularly any modifications to vascular anatomy that could have influenced blood flow patterns following PVT. The timing of PVT formation was determined based on the first imaging follow-up in which thrombosis was detected. In cases of PVTn, diagnosis was often incidental—based on findings of splenomegaly or thrombocytopenia during evaluation for unrelated illness. In selected cases, thrombophilia screening was also performed in this group.

Radiological investigations included color Doppler ultrasound, computed tomography (CT), magnetic resonance imaging (MRI), and angiography (Allura Xper FD20, Philips Healthcare). Color Doppler ultrasound was performed weekly during the first month post-diagnosis, biweekly from months two to six, and every 6–12 months thereafter. CT, MRI, or angiography was performed when ultrasound findings were inconclusive.

Endoscopic evaluation (Olympus Corp) was conducted in all children with PVT at the time of diagnosis and during follow-up at six-month intervals—or more frequently when clinically indicated. Esophageal varices were classified based on location (lower, middle, or upper esophagus) and graded as either small (grade I, <5 mm) or medium-to-large (grade II–III, >5 mm) [29]. In children under two years or weighing less than 10 kg, sclerotherapy was preferred. In all other cases, endoscopic variceal ligation was the method of choice. All procedures were performed by one of three endoscopists, each with a minimum of five years’ experience.

The approval by an ethics committee was not required due to the observational, retrospective nature of the study and the use of anonymized data. Furthermore, written informed consent was obtained for every diagnostic and interventional procedure. Moreover, authorization to access and use clinical records was obtained in accordance with institutional policies. The study complied with the international ethical guidelines for biomedical research involving human subjects, with good clinical practice guidelines, and with the Declaration of Helsinki.

### Statistical Analysis

Continuous variables were expressed as mean ± standard deviation (SD). The non-parametric Mann–Whitney U test was used to compare differences between groups (native versus transplanted livers; presence versus absence of variceal bleeding).

Categorical variables were presented as frequencies and percentages, and comparisons were made using the Chi-square (χ^2^) test or Fisher’s exact test, as appropriate.

Graphical distributions of spleen size, platelet count, and age were compared between PVTn and PVTt groups, as well as by the presence or absence of variceal bleeding, using Kernel density estimation.

## 3. Results

### 3.1. Varices and Bleeding

Clinical characteristics of the study population are summarized in Table 1. The majority of patients had biliary atresia, and no associated splenic malformations were observed. Age and sex at the time of PVT diagnosis did not differ significantly between the two groups. The mean follow-up duration was 12.97 years for PVTt and 8.89 years for PVTn. No thrombophilic disorders were identified in our cohort.

During the study period, PVTt was diagnosed in 37 out of 610 children (6%), while 36 children with PVTn formed the control group. Overall, children with PVTt tend to be younger at diagnosis (1.7 years) compared to those with PVTn (3.1 years), although the difference is not statistically significant (*p* = 0.34). Remarkably, younger children (especially under 2 years of age) tend to have a higher risk of developing PVTt due to smaller portal vein diameter, technical challenges in vascular anastomosis, lower body weight, and use of reduced grafts. 

In pediatric liver transplant recipients, esophageal varices developed within 10 years post-transplant, with a peak incidence during the first 2 years. In contrast, in children with native livers, varices typically developed within 2 years of PVT diagnosis (see Figure 1 and Figure 2).

In the PVTt group, only 2/37 (5%) patients had intrahepatic cavernous transformation, compared to 10/36 (28%) in the PVTn group (*p* < 0.001). Most cases of PVTt involved only the extrahepatic portal vein.

Overall, 27/37 (73%) of PVTt patients developed esophageal varices of any grade, with 10 of these (37%) presenting exclusively with stable grade I varices. In the PVTn group, 30/36 (83%) developed varices, of whom 7 (23%) had only grade I varices. Large esophageal varices (grade II–III) requiring eradication were found in 14/37 (37.8%) PVTt patients, compared to 23/36 (63.9%) in the PVTn group (*p* = 0.02).

Endoscopic band ligation was performed in 12/14 (85%) PVTt patients with large varices and in 15/23 (65.2%) PVTn patients (*p* = 0.17), indicating a similar choice of eradication technique across groups.

Spontaneous gastrointestinal bleeding occurred in 7/37 (19%) PVTt patients and 4/36 (11%) PVTn patients, with no statistically significant difference (*p* = 0.35). Only one patient in each group experienced bleeding before PVT diagnosis. Notably, among those who bled, 5/7 (71%) in the PVTt group had lower gastrointestinal bleeding, while all 4/4 (100%) in the PVTn group experienced upper gastrointestinal bleeding from esophageal varices.

During follow-up, additional variceal treatment was required in 4/37 (11%) PVTt patients and 8/36 (22%) PVTn patients (*p* = 0.80).

### 3.2. Risk Factors

The mean spleen length was 9.3 cm (SD 3.4) in transplanted children (PVTt) compared to 7.4 cm (SD 4.29) in non-transplanted children (PVTn), with a statistically significant difference (*p* = 0.039). The mean platelet count was 76,054 × 10^3^/L (SD 37,535) in PVTt patients and 93,916 × 10^3^/L (SD 65,585) in PVTn patients, with no significant difference between groups (*p* = 0.16).

In the PVTt group, spleen size and platelet count at the time of variceal development overlapped with the worst values recorded during follow-up. Conversely, in the PVTn group, spleen and platelet values at the time of variceal bleeding tended to cluster around the lower end of the spectrum, in comparison to their worst values observed during follow-up (Figure 3 and Figure 4).

Additionally, in children without evidence of varices, signs of hypersplenism—specifically enlarged spleen size and low platelet count—were more pronounced in the PVTt group than in the PVTn group (Figure 3 and Figure 4).

The platelet/spleen size z-score ratio was evaluated in all children with PVTt and PVTn, but did not predict the risk of variceal bleeding in either group (*p* = 0.17 and *p* = 0.7, respectively).

Among LT recipients, neither the type of transplant nor the presence of retransplantation was associated with an increased risk of developing varices (*p* = 1.00 and *p* = 0.39, respectively). Of the nine PVTt patients who underwent gastric vein ligation during transplant surgery, four developed varices. Among the 28 patients who did not have gastric vein ligation, 10 developed varices (*p* = 0.70) (see Table 2).

### 3.3. Long-Term Outcome

The therapeutic management of children with PVTt and PVTn is summarized in Table 1.

In our series, no clear age-related differences were observed in treatment or outcomes for PVT in either native liver or transplanted graft groups.

Endoscopic eradication of varices was successful in 9/37 (24%) PVTt patients and 14/36 (39%) PVTn patients.

Percutaneous recanalization with angioplasty was considered only in 8/37 (22%) PVTt patients, while no radiologic interventions were performed in the PVTn group (*p* = 0.012). TIPS was performed in 1/37 (3%) PVTt and 1/36 (3%) PVTn patients, with no significant difference between the groups (*p* = 0.98).

Surgical shunt or bypass procedures were carried out in 5/37 (13%) PVTt patients compared to 14/36 (39%) PVTn patients with a statistically significant difference (*p* = 0.013). Among the PVTt group, 2 underwent a splenorenal shunt and 3 underwent a Meso-Rex bypass. In the PVTn group, 10/14 had a splenorenal shunt, and 4/14 underwent a Meso-Rex bypass. Additionally, splenectomy was performed in 3/14 PVTn patients alongside shunt surgery.

None of the PVTn patients required liver transplantation. In contrast, 5/37 PVTt patients underwent retransplantation, although only one case listed PVT as the primary indication. The remaining indications included chronic rejection and other vascular or biliary complications.

Three deaths occurred in the PVTt group: one during bypass surgery, another during retransplantation, and a third due to acute liver failure unrelated to portal cavernoma. In the PVTn group, two patients died from causes unrelated to PVT during follow-up.

## 4. Discussion

Over the years, we have observed unexpected clinical features in PVTt compared to PVTn, which are not fully explained by current knowledge and management protocols for portal hypertension (PH). Therefore, we conducted a retrospective cohort study to evaluate the risk of developing PH with gastrointestinal bleeding and to describe the clinical, endoscopic, radiological, and surgical management of children with PVTt and PVTn. Our aim was to compare the clinical features of non-cirrhotic PVT in these two distinct settings.

In our cohort, the incidence of PVT in pediatric liver recipients was 6%, consistent with previous reports [10], with peak development of varices occurring within two years after liver transplantation (LT). It is known that the mean age at diagnosis in children with PVTn is around four years, and in 30–50% of cases, diagnosis follows gastrointestinal bleeding [2,30]. Our data are consistent with these observations: varices were typically diagnosed at approximately four years of age and within two years of PVT diagnosis. At least one episode of gastrointestinal bleeding has been reported in about 60% of children with PVTn [17]. Similarly, we found that 62% of PVTn patients developed large varices at risk of bleeding [31]. In contrast, only 38% of PVTt patients developed large varices, with a statistically significant difference between the groups.

The characteristics of gastrointestinal bleeding also differed between groups. PVTt patients exhibited a lower risk of esophageal varices but a higher prevalence of bleeding from the lower gastrointestinal tract. We hypothesize that this may be related to surgical modifications during LT, such as portal vein anastomosis, that may exclude the gastric vein from the portal circulation. However, this was not confirmed by our data. Other variables, such as transplant type or retransplantation, were not significantly associated with the development of varices. Nonetheless, we suggest that altered portal drainage due to vascular reconstruction and Roux-en-Y biliary anastomosis may contribute to bleeding from distal sites.

Interestingly, clinical hypersplenism—defined by splenomegaly and thrombocytopenia—was more pronounced in PVTt patients, despite their lower incidence of esophageal varices requiring treatment. Moreover, the platelet-to-spleen size z-score ratio did not predict variceal bleeding risk in either group, suggesting limited clinical utility in guiding surveillance.

The optimal therapeutic strategy for variceal bleeding in pediatric PH remains controversial and multifactorial, involving pharmacological, endoscopic, radiologic, and surgical options [32,33]

This complexity increases when PH presents as lower gastrointestinal bleeding, which is less amenable to endoscopic control. Endoscopic techniques, such as sclerotherapy and variceal ligation, are effective for acute variceal bleeding and variceal eradication [21,34]. In our study, fewer than half of PVTt patients required endoscopic treatment, compared to over half of PVTn patients. These findings confirm that PVTn carries a higher risk of developing large esophageal varices requiring intervention, while PVTt more often presents with distal bleeding not suitable for endoscopy. In children unresponsive to endoscopic or medical therapy, radiologic or surgical interventions may be required [25,35]. In a previous study [10], 34% of PVTn patients required surgery after a follow-up of 11.3 years. This aligns with our findings: shunt surgery was significantly more common in PVTn, while angioplasty was more frequently performed in PVTt.

Radiologically, most PVTt cases were limited to extrahepatic cavernoma, whereas PVTn more frequently involved both intra- and extrahepatic portal vein obstruction. Additionally, PVTt patients often presented with varices near the hepatico-jejunostomy site.

To our knowledge, this study represents one of the most extensive series on pediatric PVT in a high-volume liver transplant center performing 30–35 pediatric LTs per year. Our findings suggest that PVTt is a distinct clinical entity compared to PVTn. Specifically, the risk of esophageal varices is lower in PVTt, despite a higher incidence of hypersplenism. PVTt also more frequently involves the lower gastrointestinal tract, whereas PVTn primarily affects the esophageal region. These observations support the need for tailored clinical surveillance and management strategies between the two groups. We recommend routine endoscopic screening for all PVTn patients after diagnosis. In PVTt patients without gastrointestinal symptoms and low-risk profiles, endoscopy could be deferred or individualized.

This study has several limitations, primarily due to its retrospective design, which might have limited the quality and completeness of the data. Second, the limited sample size suggests caution in the interpretation of the findings. Third, the generalizability of the findings should be limited to similar patients. Prospective studies are needed to refine surveillance strategies and optimize management protocols for both PVTn and PVTt patients.

## 5. Conclusions

This study highlights distinct clinical differences between portal vein thrombosis in children with native livers and those following liver transplantation. PVTt was associated with a lower risk of esophageal varices and bleeding, but a higher incidence of hypersplenism and lower gastrointestinal bleeding. In contrast, PVTn patients were more likely to develop esophageal varices requiring endoscopic treatment. These findings support the need for tailored clinical surveillance and management strategies in these two patient populations. In particular, routine endoscopic evaluation is recommended for PVTn, while PVTt patients may benefit from individualized surveillance approaches based on clinical presentation. Prospective studies are needed to better elucidate the underlying mechanisms and to develop more targeted, effective treatment protocols.

## Figures and Tables

**Figure 1 biomedicines-13-02061-f001:**
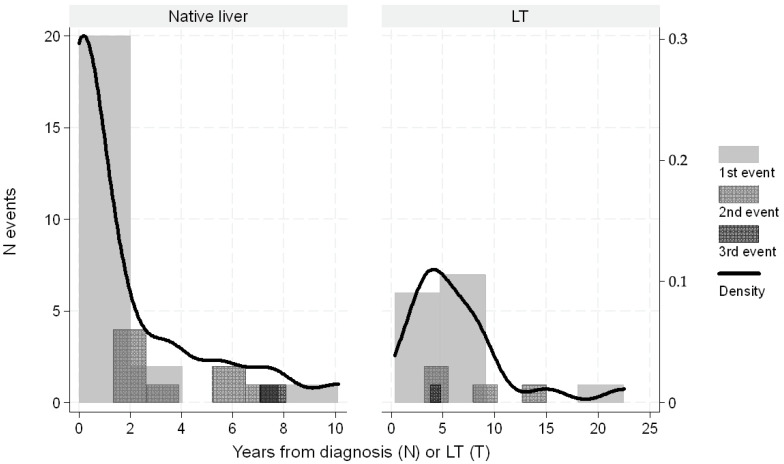
Distribution of number of varices as first or subsequent episodes by years from diagnosis or transplantation. The overlapped black line indicates the Kernel density function.

**Figure 2 biomedicines-13-02061-f002:**
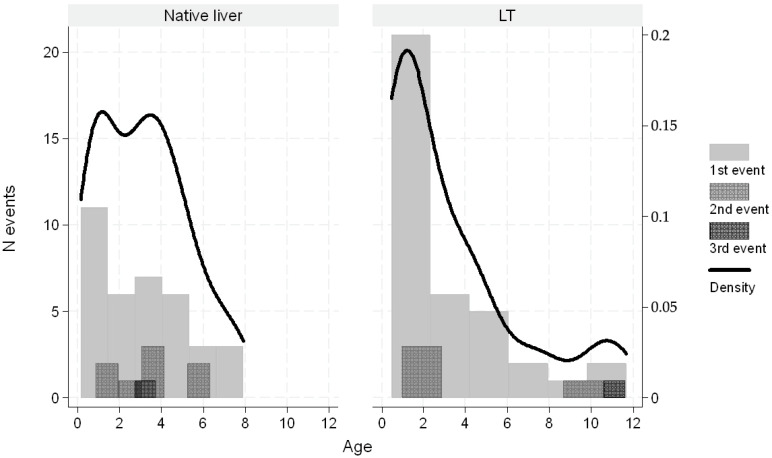
Distribution of number of varices as first or subsequent episodes by years from diagnosis or by patient age. The overlapped black line indicates the Kernel density function.

**Figure 3 biomedicines-13-02061-f003:**
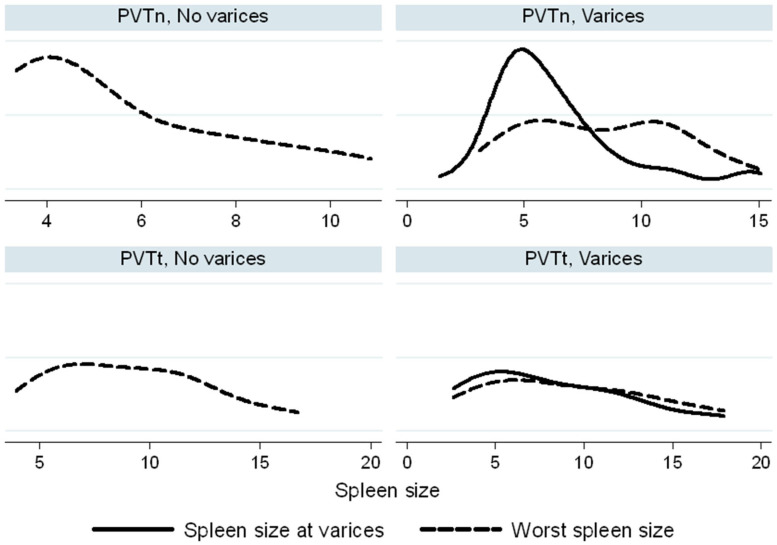
Spleen length (cm) in PVT and PVTn patients. Worst spleen measurements (dashed lines) were compared to spleen sizes at the time of varices (continuous lines). The Kernell density function is used to smooth the distribution of spleen sizes.

**Figure 4 biomedicines-13-02061-f004:**
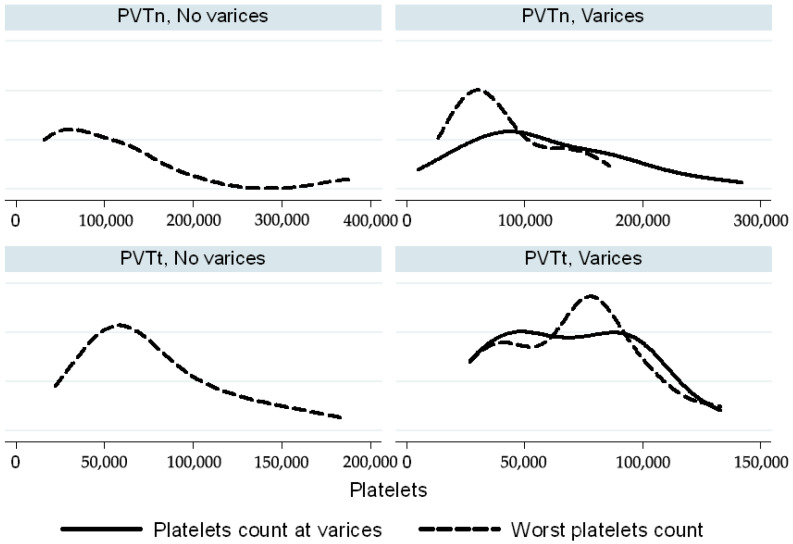
Platelet count (×10^3^/L) in PVT and PVTn patients. Worst platelet count (dashed lines) was compared to platelet count at the time of varices (continuous lines). The Kernell density function is used to smooth the distribution of platelet count.

**Table 1 biomedicines-13-02061-t001:** Features of study population.

PARAMETER	TOTAL	PVTN	PVTT	*p*-VALUE
	N = 73	N = 36	N = 37	
Sex				0.28
F	33 (45.2%)	14 (38.9%)	19 (51.4%)	
M	40 (54.8%)	22 (61.1%)	18 (48.6%)	
Age at diagnosis	2.4 (0.9–4.4)	3.1 (1.0–4.7)	1.7 (0.8–4.3)	0.34
Biliary Atresia				* <0.001
NO	44 (60.3%)	36 (100.0%)	8 (21.6%)	
YES	29 (39.7%)	0 (0.0%)	29 (78.4%)	
Large varices				* 0.026
No	36 (49.3%)	13 (36.1%)	23 (62.2%)	
Yes	37 (50.7%)	23 (63.9%)	14 (37.8%)	
Varices endoscopic treatment				0.17
Ligation + sclerosis	27 (73.0%)	15 (65.2%)	12 (85.7%)	
Sclerosis	10 (27.0%)	8 (34.8%)	2 (14.3%)	
ENLARGED Spleen length, SD				
Median, IQR (CM)	7.6 (5.2–10.9)	7.1 (4.3–10.6)	8.8 (6.0–12.3)	* 0.047
Mean, SD (CM)	8.4 (3.9)	7.4 (3.4)	9.3 (4.2)	* 0.039
LowER Platelets, McL				
Median, IQR	70,000.0 (50,000.0–107,000.0)	71,000.0 (52,000.0–126,000.0)	70,000.0 (49,000.0–94,000.0)	0.36
Mean, SD	84,863.0 (53,631.3)	93,916.7 (65,585.4)	76,054.1 (37,535.9)	0.16
Bleeding				0.35
No	62 (84.9%)	32 (88.9%)	30 (81.1%)	
Yes	11 (15.1%)	4 (11.1%)	7 (18.9%)	
Surgical shunt				* 0.013
No	54 (74.0%)	22 (61.1%)	32 (86.5%)	
Yes	19 (26.0%)	14 (38.9%)	5 (13.5%)	
Radiological treatment				* 0.012
TIPS	2 (2.7%)	1 (2.8%)	1 (2.7%)	
no	63 (86.3%)	35 (97.2%)	28 (75.7%)	
angioplasty	8 (11.0%)	0 (0.0%)	8 (21.6%)	

Legend: SD: standard deviation. * exponentiated coeffocients: *p* < 0.05.

**Table 2 biomedicines-13-02061-t002:** Risk factors of PH in PVTt.

	TOTAL	NO VARICES	VARICES	*p*-VALUE
	N = 37	N = 23	N = 14	
Type of transplant				1.00
Whole liver	1 (2.7%)	1 (4.3%)	0 (0.0%)	
Left lateral segment	36 (97.3%)	22 (95.7%)	14 (100.0%)	
Retransplant				0.39
No	30 (81.1%)	20 (87.0%)	10 (71.4%)	
Yes	7 (18.9%)	3 (13.0%)	4 (28.6%)	
Gastric vein clamp				0.70
No	28 (75.7%)	18 (78.3%)	10 (71.4%)	
Yes	9 (24.3%)	5 (21.7%)	4 (28.6%)	

## Data Availability

The original contributions presented in this study are included in the article. Further inquiries can be directed to the corresponding author.

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
