# Peer review of "Distinctive Clinical Features of Portal Hypertension in Children with Portal Vein Thrombosis Following Liver Transplantation"

_biomedicines, 2025, doi:10.3390/biomedicines13092061_

Round 1
Reviewer 1 Report
Comments and Suggestions for Authors
Strengths of the review:
- This study highlights the distinctive clinical features of portal vein thrombosis (PVT) in children with native liver transplantation (PVTn) compared with those after liver transplantation (PVTt).
- The findings suggest that post-transplant PVT has unique characteristics compared to DVT in native livers. This is associated with a lower risk of developing oesophageal varices in patients with PVTt than in those with PVTn.
- These findings suggest that clinical management and surveillance strategies for DVT should differ between these two patient populations.
- It is a study that describes and compares the clinical features of non-cirrhotic PVT in these two different settings.
Weakness:
- Materials and Methods: Did they have authorization from the clinical record or the corresponding department of the hospital? It is necessary to declare it. Were there inclusion and exclusion criteria in patient selection other than PVTt and PVTn?
- Prospective studies are needed to develop management strategies, clinical surveillance, and, above all, improve treatment protocols for patients with these conditions.
- It is suggested to include a graphical abstract that explains in detail the methodology used in this study, to make it more visually attractive.
- What does OLT mean in Figures 1 and 2? Include.

It is suggested to review the style correction, to improve the manuscript.
Author Response
Strengths of the review:
This study highlights the distinctive clinical features of portal vein thrombosis (PVT) in children with native liver transplantation (PVTn) compared with those after liver transplantation (PVTt).
The findings suggest that post-transplant PVT has unique characteristics compared to DVT in native livers. This is associated with a lower risk of developing oesophageal varices in patients with PVTt than in those with PVTn.
These findings suggest that clinical management and surveillance strategies for DVT should differ between these two patient populations.
It is a study that describes and compares the clinical features of non-cirrhotic PVT in these two different settings.
Thank you very much for your focused summary of our work.
We have addressed all your comments in the revised manuscript.
Weakness:
Materials and Methods: Did they have authorization from the clinical record or the corresponding department of the hospital? It is necessary to declare it.
Thank you very much for you comment. Written informed consent was obtained for every diagnostic and interventional procedure. This study was conducted in respect of the ethical standards laid down in the 1964 Declaration of Helsinki. Please see page 4, lines 138-142
Were there inclusion and exclusion criteria in patient selection other than PVTt and PVTn?
Thank you very much for you comment. We have added a paragraph explaining the inclusion and exclusion criteria of our study. Please see page 4, lines 110-117
Prospective studies are needed to develop management strategies, clinical surveillance, and, above all, improve treatment protocols for patients with these conditions.
Thank you very much for bringing up this point. We have added a paragraph to highlight the need for further prospective studies to improve management strategies in these patients. Please see page 12, lines 342-343
It is suggested to include a graphical abstract that explains in detail the methodology used in this study, to make it more visually attractive.
Thank you very much for your suggestion. We have created a graphical abstract to explain the methodology. Please see page 2
What does OLT mean in Figures 1 and 2?
Thank you for this comment. It was a typographical error. The acronym OLT has been changed to LT (liver transplant), please see figures 1 and 2, page 9.
Reviewer 2 Report
Comments and Suggestions for Authors
The manuscript title was “Distinctive Clinical Features of Portal Hypertension in Children with Portal Vein Thrombosis Following Liver Transplantation”. The research was about the clinical features and management of PVT in children post-transplant (PVTt) compared to 22 those with PVT in native livers (PVTn). The research was meaningful. The specific advice was as follow.
- How to intervenethis risk including the clinically and in terms of diet.
- Is it predictablefor the liver transplants (LT) leading to portal hypertension (PH) in previous?
Author Response
The manuscript title was “Distinctive Clinical Features of Portal Hypertension in Children with Portal Vein Thrombosis Following Liver Transplantation”. The research was about the clinical features and management of PVT in children post-transplant (PVTt) compared to 22 those with PVT in native livers (PVTn). The research was meaningful. The specific advice was as follow.
Thank you very much for your kind feedback.
We have addressed all revisions in the revised manuscript.
How to intervene this risk including the clinically and in terms of diet.
Thank you very much for your comment. We have added a paragraph about the role of endoscopy in PVTt and PVTn. Please see page 12, lines 334-338
Although, portal hypertension and PVT can lead to malnutrition and nutritional management is essential, there are no specific dietary suggestions recommended by the current literature, unless there is associated cholestasis. For this reason this aspect has not been investigated.
Is it predictable for the liver transplants (LT) leading to portal hypertension (PH) in previous?
Thank you for this interesting point. While some risk factors for developing PH after LT (especially secondary to PVTt) are known, there is no validated predictive model that reliably identifies pediatric LT candidates who will develop clinically significant PH. Please see page 3, lines 89-92
Reviewer 3 Report
Comments and Suggestions for Authors
Dear colleagues!
I read with interest your manuscript "Distinctive Clinical Features of Portal Hypertension in Children with Portal Vein Thrombosis Following Liver Transplantation" submitted to Biomedicines. The paper is based on retrospective observational comparative trial results aimed to describe clinical features and management of portal vein thrombosis (PVT) in children after liver transplantation (PVTt) compared to those with PVT in native livers (PVTn). The study brings new to the field, fills in knowledge gap and may be interesting for the readers.
I have a few minor comments / suggestions.
Please, add more details on the type / power of the hospital including number of transplantations performed annually.
Please, describe search criteria for the cases (taking to the account retrospective nature of the study).
Tables: please, provide data in SI format, if possible.
Discussion: please, describe strengths and limitations of the study.
Please, provide your vision / suggestions based on the results obtained (when to follow-up, at which groups), even despite this should be confirmed in the future research. Can you suggest any prophylaxis for the main group?
Conclusions: please, add some confirmative values for the statement based on the results obtained.
My comments in no way diminish the dignity of the work. I hope that they help you to make the manuscript even better
Comments on the Quality of English LanguageSome abbreviations are not disclosed.
The paper requires minor language polishing.
Author Response
Thank you very much for your focused summary of our work.
We have addressed all your comments in the revised manuscript.
Please, add more details on the type / power of the hospital including number of transplantations performed annually.
Thank you very much for your comment. We have added a paragraph describing
the power of our pediatric LT Center, please see page 11 lines 330-331
Please, describe search criteria for the cases (taking to the account retrospective nature of the study).
Thank you very much for your comment. We have added a paragraph regarding inclusion/exclusion criteria, please see page 4, lines 110-117. Furthermore we added a graphical abstract to better explain the methodology of our study, please see page 2
Tables: please, provide data in SI format, if possible.
Thank you very much for your comment. We have modified our data according to SI.
Discussion: please, describe strengths and limitations of the study.
Thank you very much for your comment. We have added a paragraph regarding the strengths and the limitations of our study at the end of the discussion. Please see page 11, lines 330-331, 339-342.
Please, provide your vision / suggestions based on the results obtained (when to follow-up, at which groups), even despite this should be confirmed in the future research. Can you suggest any prophylaxis for the main group?
Thank you very much for bringing up this important point. A paragraph was added to outline our findings and to suggest a potential surveillance algorithm. Please see page 12 lines 334-338
Unlike in adults, primary prophylaxis with beta blockers is not routinely recommended in children in agreement with our experience. Given that, we do not recommend any pharmacological prophylaxis.
Round 2
Reviewer 2 Report
Comments and Suggestions for Authors
None. The manuscript can be revised appropriate.
Author Response
Thank you very much